# Human centromere repositioning activates transcription and opens chromatin fibre structure

Catherine Naughton[1], Covadonga Huidobro [1], Claudia R. Catacchio[1,2], Adam Buckle[1], Graeme R. Grimes[1], Ryu-Suke Nozawa[1], Stefania Purgato[1,3], Mariano Rocchi[2] & Nick Gilbert [1] ✉

Human centromeres appear as constrictions on mitotic chromosomes and form a platform for kinetochore assembly in mitosis. Biophysical experiments led to a suggestion that repetitive DNA at centromeric regions form a compact scaffold necessary for function, but this was revised when neocentromeres were discovered on non-repetitive DNA. To test whether centromeres have a special chromatin structure we have analysed the architecture of a neocentromere. Centromere repositioning is accompanied by RNA polymerase II recruitment and active transcription to form a decompacted, negatively supercoiled domain enriched in 'open' chromatin fibres. In contrast, centromerisation causes a spreading of repressive epigenetic marks to surrounding regions, delimited by H3K27me3 polycomb boundaries and divergent genes. This flanking domain is transcriptionally silent and partially remodelled to form 'compact' chromatin, similar to satellite-containing DNA sequences, and exhibits genomic instability. We suggest transcription disrupts chromatin to provide a foundation for kinetochore formation whilst compact pericentromeric heterochromatin generates mechanical rigidity.

Centromeres are highly specialised genomic loci necessary to maintain genome stability. Cytogenetically they are the primary constriction of a metaphase chromosome and functionally provide an assembly site for the kinetochore, a multiprotein structure that forms attachments to the microtubules of the mitotic and meiotic spindles[1]. In mitosis the kinetochore is composed of a trilaminar structure with an outer layer binding to microtubules, but the architecture of the underlying chromatin fibre is unknown[2].

Human centromeric chromatin is assembled from CENP-A nucleosomes[3] and repetitive α-satellite DNA sequences that span 250–5000 Kb[4,5]. Mouse acrocentric chromosomes have a similar organisation but, in this case, a small centromeric domain of minor satellite is flanked by a larger region of major satellite, which in interphase coalesces to form large dense chromocentres, enriched in heterochromatic marks and HP1 protein[6]. A prevailing hypothesis is that repetitive satellite sequences at centromeres form compact heterochromatin, which provides a stable scaffold for the kinetochore[7]. This idea is supported by biophysical experiments: (i) analysis of satellite containing mouse and human centromeric chromatin by sucrose gradient sedimentation shows that it sediments more rapidly than expected for its size, indicating that is has a compact chromatin structure, analogous to a rigid rod[8]; (ii) in different species nucleosomes are positioned regularly on satellite sequences consistent with the assembly of chromatin fibres having a regular and stable structure[9,10]; (iii) in vitro pulling experiments indicate that regularly folded chromatin has the biophysical properties of a stiff spring[11].

In contrast to biophysical data that indicates satellite containing centromeric chromatin has a uniform compact architecture, immunofluorescence analysis on extended interphase chromatin fibres[12–14] show that it is divided into core and pericentromeric domains. The

[1]MRC Human Genetics Unit, The University of Edinburgh, Crewe Rd, Edinburgh EH4 2XU, UK. [2]Department of Biology, University of Bari, Via Orabona 4, 70125 Bari, Italy. [3]Department of Pharmacy and Biotechnology, University of Bologna, 40127 Bologna, Italy. ✉e-mail: Nick.Gilbert@ed.ac.uk

centromere core domain is enriched in active histone modifications indicative of transcription, whilst the surrounding pericentromeric regions are marked by repressive histone marks[13]. This core centromeric transcription is essential for proper centromere function and identity[15–27], with core centromere identity being epigenetically defined by the variant histone CENP-A. CENP-A interacts with CENP-C through the LEEGLG motif at the extreme C terminus[28] and RNA[16,18] to form an anchor for kinetochore formation, whilst the pericentromere recruits cohesin and condensin to regulate chromatin stiffness[29]. Whilst mechanisms for CENP-A recruitment are slightly different between species (for example budding yeast, S. cerevisiae versus the nematode, C. elegans versus fission yeast, S. pombe and higher eukaryotes[30]), it appears that a function of transcription at the core is to facilitate the incorporation of CENP-A containing nucleosomes[23]. Furthermore, studies in S. pombe indicate that chromatin remodelers spread from the centromeric core to surrounding pericentromeric regions[31]. This two-domain organisation appears critical for centromere stability, as experiments disrupting either transcription levels, or heterochromatic marks, affect chromatin compaction and result in mitotic defects[32–37].

Higher eukaryotic centromeres are typically located on repetitive DNA sequences, but they can also be found at euchromatic sequences[38]. These neocentromeres often form in response to chromosome instability in cancer which deletes the canonical centromere, but can also occur congenitally. Neocentromeres that have been inherited through generations and become fixed in the population are described as evolutionary new centromeres (ENC's) and have often been inferred from studying the chromosome architecture between similarly related species[39]. Historically molecular analyses of centromeres have been challenging due to the repetitive nature of the underlying DNA, however, as neocentromeres are formed on unique DNA sequences, they provide a useful model to interrogate centromeric chromatin structure and provide insight into the properties of canonical centromeres. For example, examination of ENCs imply α-satellite DNA may be acquired over time at neocentromeres as they mature[40–42], neocentromeres lacking pericentromeric heterochromatin in cis may establish interactions to distal heterochromatin in trans[43], neocentromere formation promotes H3K9me3 loss and RNA polymerase II accumulation at the CENP-A core[44], suggesting that chromatin is remodelled to accommodate a functional centromere.

To reconcile results from biophysical and imaging-based studies we have used a neocentromere as a model system to determine whether centromeres have a special chromatin structure. Centromere repositioning is accompanied by epigenetic and chromain fibre remodelling: the CENP-A defined core becomes enriched in active epigenetic marks, RNA polymerase, and negatively supercoiled DNA, consistent with transcription. To examine the biophysical properties of the chromatin fibre, sedimentation analysis shows that it has a transcription-dependent disrupted chromatin fibre structure. These structural changes of core centromeric chromatin further affect the large-scale chromatin fibre folding of this region which becomes decompacted in a transcription-dependent manner. Strikingly, there is pronounced epigenetic remodelling and transcriptional silencing of a large 5 Mb region surrounding the centromeric core. Although there is no concomitant change in nucleosome positioning at the centromere there is evidence for partial remodelling of the flanking pericentromeric heterochromatin to form 'compact' chromatin. As this region is genomically unstable we propose that further remodelling of the pericentromeric region to form compact heterochromatin occurs as the neocentromere matures. Overall, our data indicates that centromeres are remodelled to have a special chromatin structure: chromatin fibres at the centromere core have a disrupted structure that we suggest provides a suitable foundation to attach the kinetochore components whilst flanking sequences form a compact heterochromatin-like structure that has mechanical rigidity.

## Results

### Epigenetic remodelling at a human neocentromere

To understand how new centromeres are accommodated in chromosomes and to investigate whether centromeres have a special chromatin structure required to form a stable kinetochore we used a previously identified neocentromere at 3q24 as a model system[45]. This neocentromere has been propagated across multiple generations, indicating it is stable through the germline, and is located in the vicinity of two genes but within a relatively gene-poor segment of the genome. As the parental lymphoblastoid cells are heterozygous for the neocentromere at 3q24 the chromosome harbouring the neocentromere, Neo3, was genetically isolated from the normal counterpart in a human-hamster hybrid cell line (HybNeo3; Fig. 1A) and compared to a human-hamster hybrid cell line, GM10253A, which has a single normal human chromosome 3, termed HSA3. No genetic changes were apparent in Neo3 and there was no evidence for repetitive DNA at 3q24 by deep sequencing (x30 coverage/base), whilst as reported for other neocentromeres[46–48] α-satellite persisted at the original centromere location (Fig. 1A).

The position of the neocentromere was confirmed by using DNA fluorescence in situ hybridisation (FISH) with probes to 3q24 (Supplementary Fig. 1A–C) and high-resolution mapped using ChIP with antibodies to CENP-A and CENP-C in the parental cells (Supplementary Fig. 1D), revealing a centromere core domain of 130 kb, similar to other synthetically derived neocentromeres[44]. In the derivatized human-hamster HybNeo3 cell line the centromere had drifted ≈30 kb away from the telomere and had spread to encompass a ≈190 kb domain. Centromere drift is apparent in horse and fission yeast[49,50], and may represent a natural event controlled in part by the constitutive centromere-associated network (CCAN) and buffered by repetitive satellite DNA[51].

We set out to investigate how the chromatin fibre is remodelled in response to centromere repositioning and reasoned there could be two distinct possibilities (i) neocentromeres form at a genomic location that already has the features required for centromere function or (ii) neocentromeres have the capacity to remodel the local epigenetic environment. To discriminate between these two scenarios we compared the epigenetic repertoire of 3q24 using ChIP for active (H3K27ac, H3K4me2, H4K20me1) and repressive (H3K9me3, H3K9me2) epigenetic marks. These marks were mapped via ChIP-chip (Custom 180 K Agilent microarrays designed to the 7 Mb spanning the neocentromere domain) as preliminary experiments showed ChIP-chip performed better than ChIP-seq in the human-hamster hybrid cell lines. One reason for this could be a ChIP-seq sequence dept issue due to the presence of all the hamster chromosomes in the hybrid cell lines. A small block of GC-rich DNA in the vicinity of the neocentromere amplified aberrantly and was blacklisted (Supplementary Fig. 1E).

The canonical 3q24 locus on HSA3 was decorated with active marks (Fig. 1B, C) coincident with actively transcribed genes in a euchromatic region, whilst repressive heterochromatic marks were absent. In contrast, after neocentromerisation, a large 5 Mb heterochromatin domain marked by H3K9me2/3 formed around the centromere on Neo3 (Fig. 1B, yellow box). Focal active marks in the vicinity of genes were absent and there was a significant loss of H3K27ac, a marker of CBP/P300 activity[52]. The upstream and downstream pericentromeric regions had the epigenetic hallmarks of heterochromatin consistent with the idea that centromeres are remodelled into a repressive state, even in the absence of repetitive DNA, demonstrating that special DNA sequences are not required for heterochromatin formation. At the centromeric core (Fig. 1B, C, blue box), coincident with CENP-C binding, the chromatin was remodelled to a state distinct from the flanking pericentromeric domains, devoid of heterochromatin marks and enriched for H4K20me1. This data suggests that instead of neocentromeres adopting the local epigenetic landscape, they can remodel the local chromatin environment[44] to form distinct

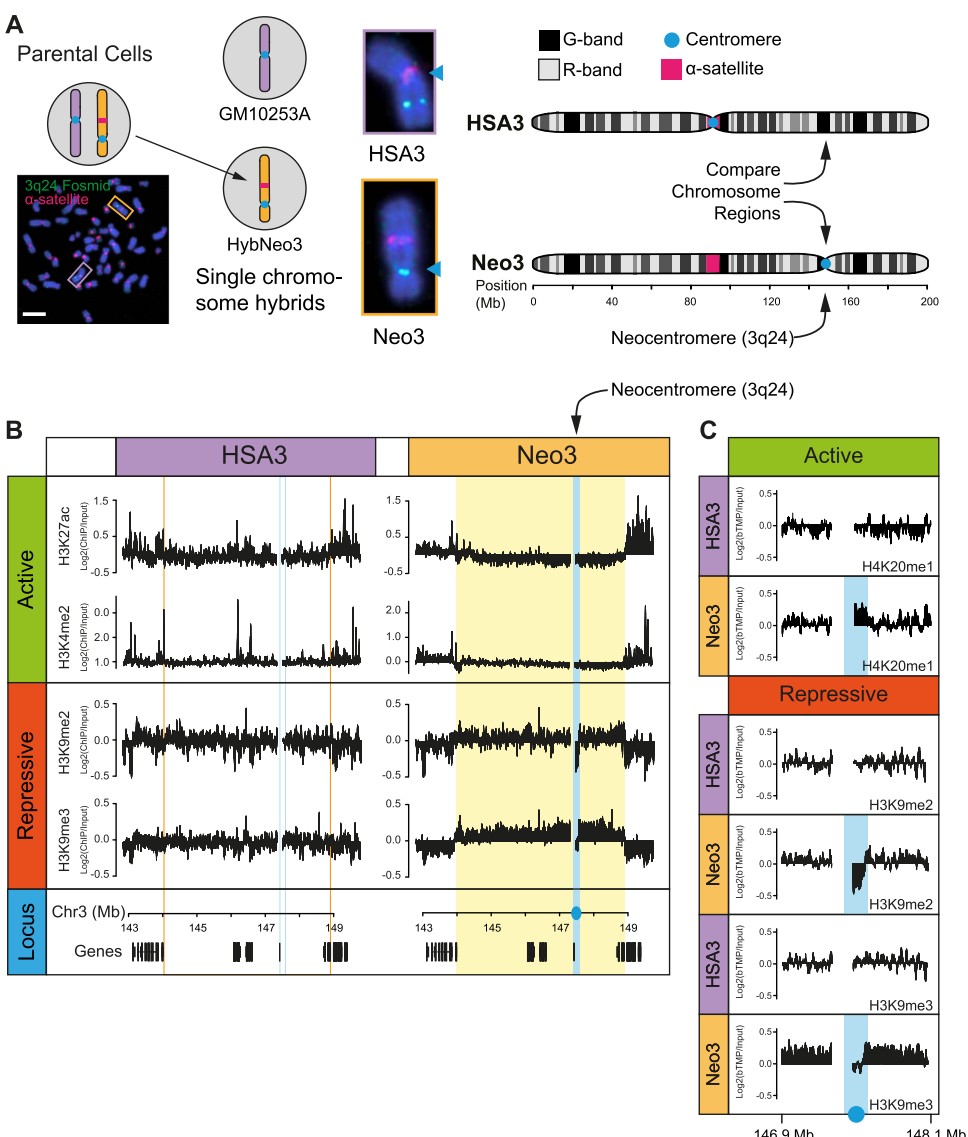

**Fig. 1 | Epigenetic remodelling after centromere repositioning. A** Schematic detailing experimental model system. Left; human lymphoblastoid cells (Parental) harbouring a canonical human chromosome 3 (HSA3; purple frame) and a chromosome 3 with a neocentromere located at 3q24 (Neo3; orange frame) were genetically manipulated to isolate Neo3 in a human-hamster hybrid (HybNeo3) for comparison to a human-hamster hybrid (GM10253A) with a single HSA3 chromosome. Middle, DNA FISH with α-satellite specific probe (red) and a fosmid probe (green) for 3q24. The primary constriction (arrow) coincides with the α-satellite array in HSA3 but is located at 3q24 in Neo3. Chromosomes were counterstained with DAPI. The bar is 5 μm. Right, chromosome 3 ideogram indicating the 3q24 chromosome region that was compared between the HSA3 and Neo3. **B** Distribution of active (H3K4me2, H3K27ac) and repressive (H3K9me3, H3K9me2) epigenetic marks measured by ChIP-chip at the 3q24 neocentromere region on HSA3 and Neo3 chromosomes. Neocentromere is marked in solid blue, equivalent position in HSA3 is marked in open blue box. Remodelled pericentromeric heterochromatin domain is marked in solid yellow, with the equivalent region on HSA3 marked in open orange box. Bottom, the position of genes at 3q24 locus. **C** Detailed view of active (H4K20me1) and repressive (H3K9me2/3) epigenetic marks surrounding the neocentromere.

centromeric and pericentromeric domains (Fig. 1B, blue and yellow boxes, respectively).

## Transcriptional landscape at a neocentromere

To understand the molecular basis for the distinct centromeric and pericentromeric domains, patterns of transcription were examined. By RT-qPCR active genes at 3q24 on HSA3 were all silenced upon centromere repositioning, as far as the distal *DIPK2A* and *HLTF* genes (Supplementary Fig. 2), suggestive of a spreading activity emanating from the centromere core domain. RNA sequencing was used to further explore the landscape and showed transcriptional repression over a 5 Mb pericentromeric domain on Neo3 (Fig. 2A). Recent data has indicated that (neo)centromeres are transcriptionally active in mitosis[22,44] and in interphase in model organisms[15–20,23,27,31,53,54]. ChIP for RNA polymerase II showed it was absent from the pericentromeric domain in Neo3, but statistically significant levels of polymerase were apparent at the centromeric core in both interphase (Fig. 2B) and metaphase (Fig. 2C) cells. However, no transcripts were detected within the centromeric core domain on Neo3 even after exosome knockdown (Fig. 2D, E). Similarly, sequencing for RNA transcripts or nascent transcripts (see methods) could not identify specific RNA species. This led us to speculate that transcription was at a very low level and dispersed across the centromeric core domain with multiple transcription initiation sites, but sufficient to remodel the local chromatin landscape.

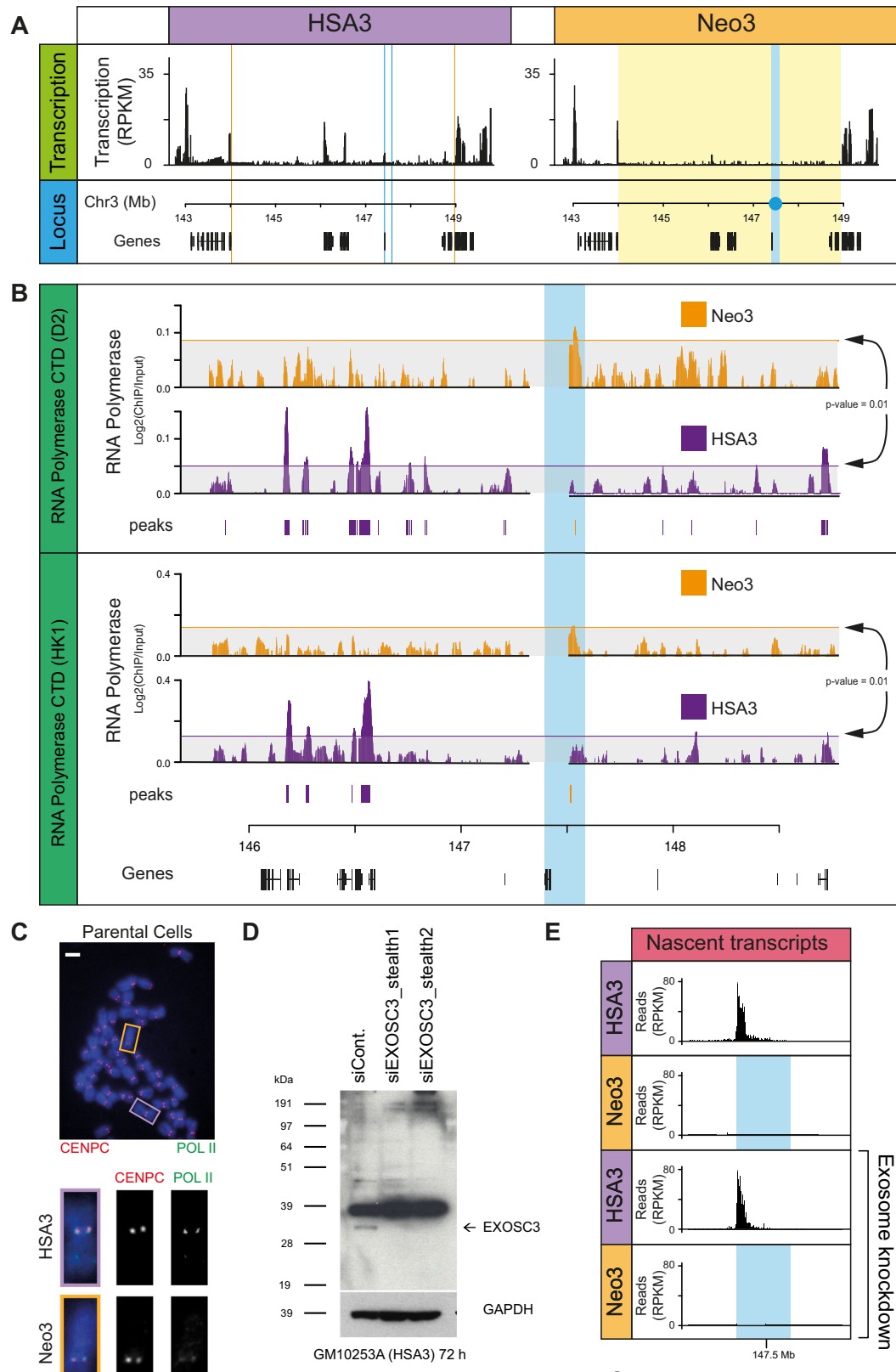

### Distinct pericentromeric domain boundaries

Centromerisation triggered epigenetic remodelling to form a repressive pericentromeric domain (Fig. 1B). To characterise how the domain was delimited, we examined facultative epigenetic marks across the locus and identified strong H3K27me3 enrichment at the telomeric end of the domain and a weak enrichment for H3K36me3 at the other end (Fig. 3A). H3K27me3 is a mark indicative of polycomb activity, whilst H3K36me3 is added by Set2 at active genes but is also reported as a facultative mark at heterochromatin[55]. These features suggest that the repressive pericentromere activity spreads until it reaches these boundaries, but raises the question as to what defines them? CTCF is an abundant protein associated with

**Fig. 2 | RNA pol II binding at a functioning human centromere. A** Normalised transcription across HSA3 and Neo3 analysed by RNA-seq. **B** Distribution of RNA pol II binding at 3q24 for Neo3 (orange) and HSA3 (purple) chromosomes. Top, RNA pol II CTD antibody (D2, Diagenode), bottom, RNA pol II CTD antibody from Hiroshi Kimura (HK1). Horizontal line (arrows) corresponds to a random permutation analysis with $p = 0.01$. Significant peaks ($p < 0.001$) called using the Ringo package for Neo3 (orange) and HSA3 (purple) are shown below the tracks. **C** Representative image of RNA pol II (CTD domain, Diagenode) (green) and CENP-

C[83] (red) immunofluorescence staining on a metaphase spread from parental cells (HSA3, purple; Neo3, orange) ($n = 3$ biologically independent experiments). Bar is 5 μm. **D** Western blot confirming EXOSC3 protein (arrow) knockdown following 72 h RNAi treatment in GM10253A cells ($n = 2$ biologically independent experiments). **E** Distribution of nascent transcripts (normalised RPKM) in 600 kb window around neocentromere (blue), mapped using TT-seq. The top panels correspond to RNAi control, bottom panels are for exosome RNAi knockdown.

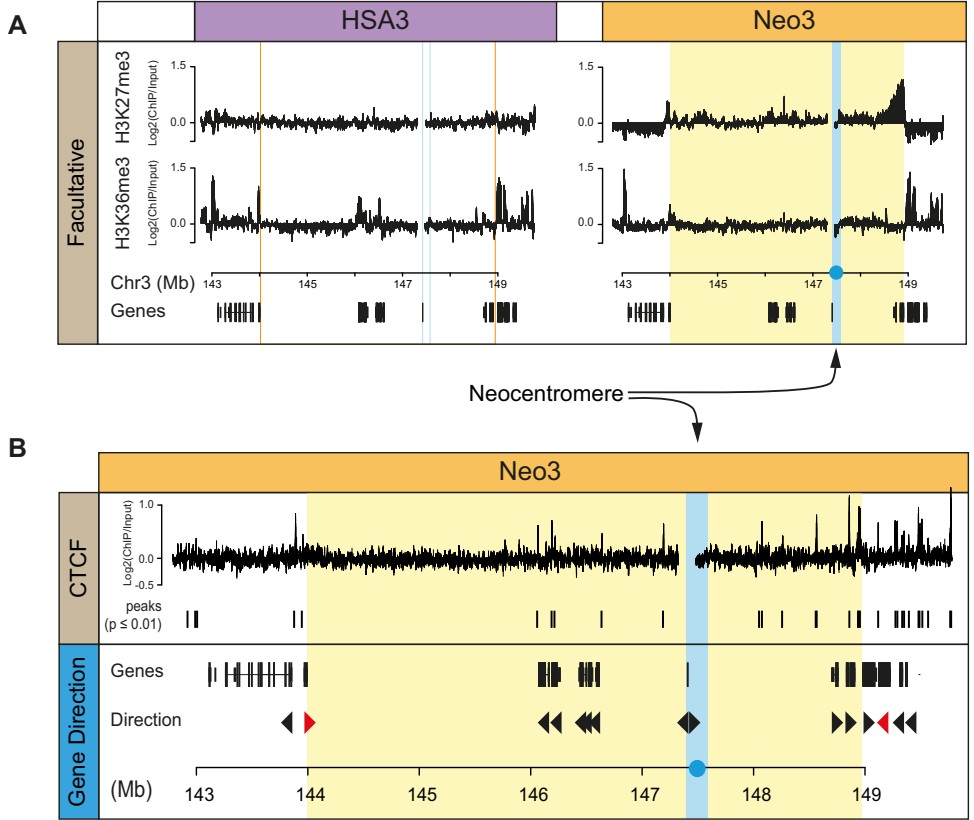

**Fig. 3 | Facultative heterochromatin and convergent genes mark transcriptionally silenced pericentromeric domain. A** Distribution of facultative heterochromatin (H3K27me3 and H3K36me3) marks measured by ChIP-chip on HSA3 and Neo3 chromosomes. **B** Top, ChIP-chip showing the distribution of CTCF at pericentromeric boundaries on Neo3 chromosomes. Bottom, schematic showing gene orientation at 3q24, convergent gene boundary marked in red. The vertical blue line corresponds to the neocentromere and repressive pericentromeric heterochromatin domain in yellow.

GC-rich DNA sequences and provides boundary activity for marking chromatin interaction domains measured by 3 C techniques[56,57]. Although CTCF was present throughout the pericentromeric domain, there were strong peaks located at both ends of the region (Fig. 3B). More strikingly and consistent with a recent study in budding yeast[58] the pericentromeric domain was also delimited by the first convergently transcribed gene encountered moving away from the centromere (Fig. 3B). These results reveal a pronounced two domain structure at centromeres with a core and pericentromeric domain flanked by boundary sites defined by convergent genes and CTCF binding.

### Local chromatin fibre remodelling after centromere repositioning

In gene-rich euchromatin nucleosomal DNA is packaged into chromatin fibres which have a disrupted or 'open' configuration[59], a structure that is particularly conspicuous at transcription start sites[60]. In contrast, the specialised chromatin found at centromeres is formed from alpha satellite[4] and CENP-A containing nucleosomes[3]. Alpha satellite makes up 3% of the human genome[61] and positions nucleosomes precisely in vivo[9] and in vitro[10]. Sedimentation studies to

examine the biophysical properties of satellite containing chromatin indicate that it has the characteristics of a rigid rod-like particle which may enable it to fold into an ordered or crystalline array[8,62]. To establish the biophysical properties of centromeric chromatin (Fig. 4A), soluble chromatin was prepared from nuclei containing HSA3 and Neo3 chromosomes and fractionated by sucrose gradient sedimentation and pulsed-field gel electrophoresis (PFGE)[59] (Supplementary Fig. 3A). Subsequently DNA corresponding to 'open' or disrupted chromatin was isolated from the agarose gel (Supplementary Fig. 3B) and used to map the chromatin fibre structure in the centromeric domain. Chromatin fibres located at the neocentromere were substantially remodelled to have a pronounced 'open' configuration (Fig. 4B) which was restricted to the CENP-C containing core, and is similar to the characteristics observed at transcription start sites[60]. Due to the presence of low-level RNA polymerase in the centromeric domain (Fig. 2B), we speculated that chromatin remodelling was linked to transcription, as has been observed in model organisms[23,31,53]. Concomitantly transcription inhibition completely abrogated the formation of disrupted chromatin fibres (Fig. 4B), demonstrating that centromeric chromatin is remodelled to have a transcription-dependent 'open' structure.

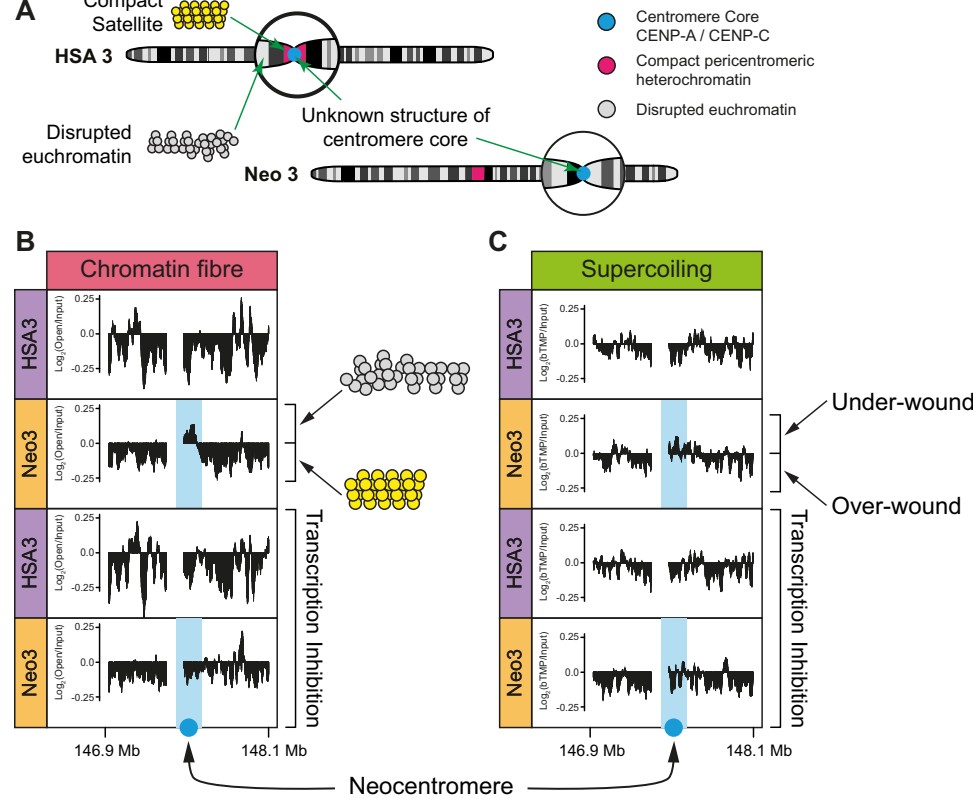

**Fig. 4 | Centromeric chromatin has a transcription-dependent underwound and disrupted chromatin fibre structure. A** Top, schematic indicating the lack of understanding of chromatin structure at canonical centromeric chromatin (HSA3) and after centromerisation (Neo3). **B** DNA corresponding to 'open' or disrupted chromatin was isolated by sucrose gradient fractionation of chromatin followed by PFGE and used to map the chromatin fibre structure in the centromeric domain.

The distribution of disrupted chromatin fibres across a 1.2 Mb region on HSA3 and Neo3, in the presence or absence of transcription inhibition (5 h α-amanitin treatment) is shown as Log2 ratio of open: input hybridisation signal aligned to the DNA sequence (Mb). **C** Organisation of negative (under-wound) and positive (over-wound) supercoils mapped by bTMP binding before or after transcription inhibition (5 h α-amanitin treatment).

Previous studies have suggested that CENP-A containing chromatin is folded differently and this could be linked to DNA supercoiling[63]. If DNA is twisted in a right-handed direction, it becomes over-wound (positive supercoiling) whilst twisting in the opposite direction, it adopts an under-wound (negatively supercoiled) configuration[64]. Our earlier work showed that the level of supercoiling is transcription dependent[65] so we hypothesised that low-level RNA polymerase II activity could impact the local DNA configuration. Using biotinylated 4,5,8-trimethylpsoralen (bTMP) as a DNA structure probe[65] centromeric chromatin was found to be enriched in negatively supercoiled DNA (Fig. 4C), in a transcription-dependent manner. These data further indicate that RNA polymerase is not present as a static component but is engaged in active transcription that remodels DNA and local chromatin fibre structure.

### Centromere repositioning is not accompanied by nucleosome repositioning

Satellite-containing pericentromeric chromatin fibres found at canonical centromeres have a rigid rod-like structure[8] that may facilitate kinetochore formation and increase the fidelity of chromosome segregation. In contrast, neocentromeres do not have repetitive α-satellite DNA sequences[46–48] to precisely position nucleosomes, so it was important to ask if centromerisation, per se, affected nucleosome positioning within the core or flanking DNA sequences. Mono and di-nucleosome fragments prepared from HSA3- and Neo3-containing nuclei using DFF nuclease (Supplementary Fig. 4A, B) were selected using biotinylated baits (Supplementary Fig. 4C) and deep sequenced. After centromerisation the size of the nucleosomal fragments did not change, despite centromeric nucleosomes being enriched in CENP-A

(Supplementary Fig. 4D–F) and no difference in nucleosome positioning (Fig. 5A) or periodicity (Supplementary Fig. 4G) was observed.

Despite no apparent change in the nucleosomal arrangement we speculated that over time pericentromeric chromatin may be remodelled to adopt a more compact configuration (scenario 2; Fig. 5B), analogous to the structure observed at canonical pericentromeres[8]. Consistent with this idea, a 250 kb region at the pericentromeric boundary had a compact chromatin fibre structure (Fig. 5B) coincident with H3K36me3 (Fig. 3A), a mark that has previously been observed at constitutive and facultative heterochromatin[55]. This indicates that H3K9me2/3 heterochromatin marks are not sufficient to generate a compact chromatin fibre structure, and pericentromeric chromatin can be remodelled to form a structure analogous to canonical satellite-containing pericentromeric chromatin.

### Decompacted large-scale centromeric chromatin

As previous studies indicated an inter-relationship between different levels of chromatin organisation[59,60,65], we speculated that after centromere repositioning local changes in chromatin structure might be propagated and influence large-scale chromatin compaction[66]. To directly test this hypothesis 3D DNA FISH, with pairs of differentially labelled fosmid probes ≈300 kb apart (Supplementary Table 1), was used to ascertain large-scale chromatin compaction at the core and pericentromeric chromatin domains (Fig. 6A). In the hamster-hybrid cells harbouring HSA3 and Neo3 chromosomes there was no apparent change in compaction in pericentromeric regions but a pronounced decompaction at the centromeric core (Fig. 6B and Supplementary Fig. 5A, B). To ensure this difference was not a consequence of comparing different cell lines the analysis was repeated in the parental cells

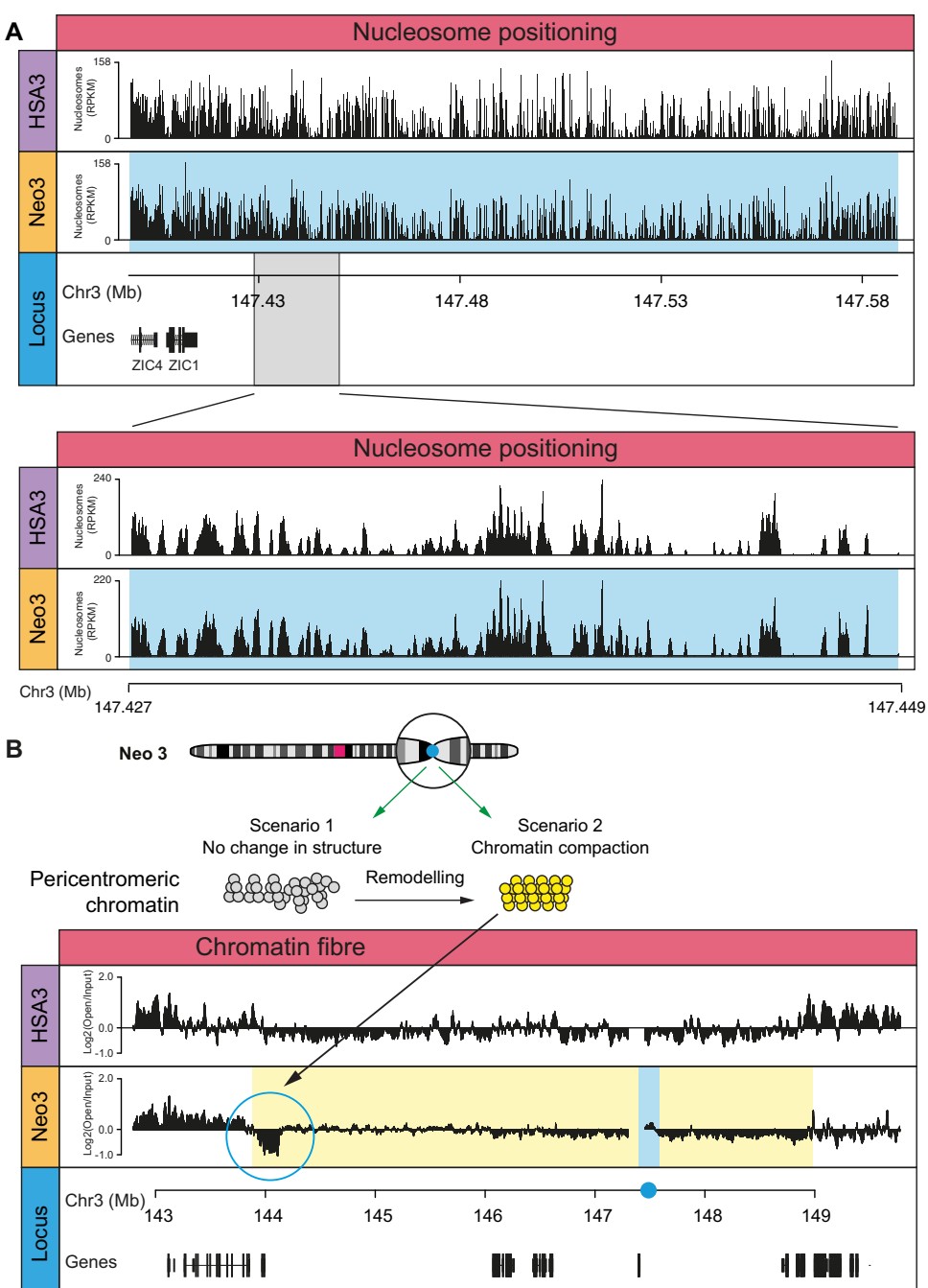

**Fig. 5 | Compact heterochromatin in pericentromeric domain. A** Nucleosome dyad coverage on the HSA3 and Neo3 chromosome at the neocentromere domain (blue) with enlarged 22 Kb region (below). **B** Top, diagram showing potential scenarios for chromatin remodelling at pericentromere after centromerisation. Bottom, DNA corresponding to 'open' or disrupted chromatin was isolated by sucrose gradient fractionation of chromatin followed by PFGE and used to map the chromatin fibre structure in the centromeric domain. The distribution of disrupted chromatin fibres across the 7MB region of HSA3 and Neo3 is shown as Log2 ratio of open:input hybridisation signal aligned to the DNA sequence (Mb). The blue bar corresponds to neocentromere and the yellow domain marks the silenced pericentromeric region.

using CENP-C immuno-FISH to discriminate between the Neo3 and HSA3 chromosomes. This similarly revealed a significant large-scale chromatin decompaction after centromerisation (Supplementary Fig. 5C, D), showing that remodelling occurs at multiple levels of centromere organisation. Large-scale chromation decompaction after centromerisation was further confirmed at another neocentromere on human chromosome 6 (Neo6) (Supplementary Fig. 6). Centromeric large-scale chromatin structure was also transcription dependent, with both α-amanitin and flavopiridol treatment causing chromatin compaction (Fig. 6C; Supplementary Fig. 5E). As bleomycin treatment

(introduces nicks) also caused large-scale chromatin to compact (Fig. 6C), this suggested the fibres were under topological strain, consistent with being negatively supercoiled (Fig. 4C). Although we were unable to find evidence for centromeric derived transcripts (Fig. 2E) we speculated that transcripts may act locally to impact chromatin structure[67]. Consistent with this idea, RNase H treatment (hydrolyses RNA in the context of a DNA/RNA hybrid) compacted centromeric but not pericentromeric chromatin structure (Fig. 6D) suggesting that a transcription-dependent RNA component stabilised decompacted centromeric chromatin.

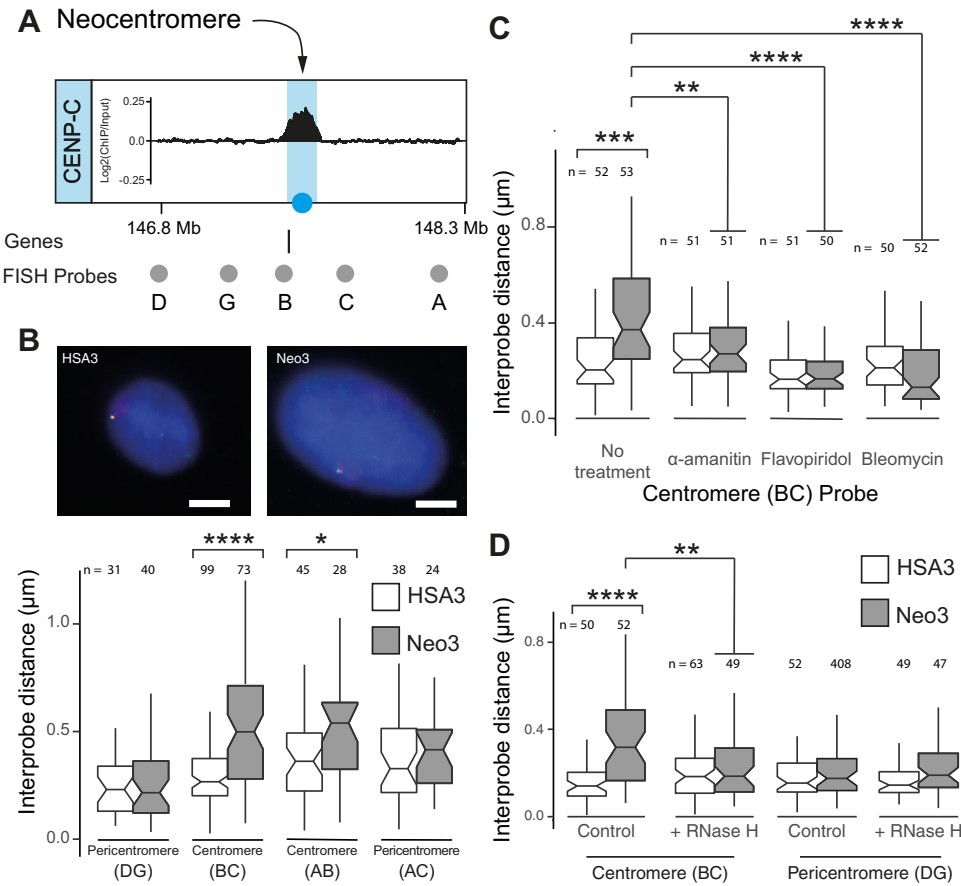

**Fig. 6 | Large-scale neocentromere chromatin fibre decompaction. A** Diagram showing fosmid FISH probes (grey circles) surrounding the CENPC-marked centromeric core domain (blue). **B** Top, representative images of 3D-FISH on HSA3 and Neo3 chromosomes hybridised to probe B (red) and C (green) in single chromosome human-hamster hybrid nuclei, counterstained with DAPI. Bar is 5 μm. Bottom, boxplot showing interprobe distance measurements (μm) for pairs of fosmid probes. Exact *p* values for a Wilcoxon test were (BC) 3.29e−01 and (AB) 0.02505. **C** Boxplot showing interprobe distance (μm) between the BC (centromere) fosmid probes in HSA3 (white) and Neo3 (grey) chromosomes in GM10253A and HybNeo3 cell lines respectively, after treatment with α-amanitin (5 h), flavopiridol (3 h) or

bleomycin (10 min). Exact *p* values for a Wilcoxon test were (no treatment) 7.073e−05, (amanitin) 0.0106, (flavopiridol) 3.69e-07 and (bleomycin) 3.6e−07. **D** Boxplot showing interprobe distance (μm) for the BC (centromere) and DG (pericentromere) pairs of fosmid probes on HSA3 or Neo3 chromosomes following RNase H treatment. Exact *p* values for a Wilcoxon test were (control) 3.616e−01 and (RNaseH) 0.01031. Source data are provided as a Source Data file. **B–D** Data are shown as boxplots with median (middle line), 25−75th percentiles (box), and min-max values (whiskers), *p*-values are for a Wilcoxon test (two-sided); *, *p* < 0.05; **, *p* < 0.01; ***, *p* < 0.001; ****, *p* < 0.0001. Source data are provided as a Source Data file.

### Inherent genome instability at a neocentromere

Whilst neocentromeres form fully functional kinetochores that are stably propagated[68], they are still associated with higher chromosome mis-segregation rates and mitotic errors[69,70]. To quantify neocentromere stability, we examined the chromosome architecture and copy number of the centromeric and pericentromeric domains in cells propagated for different amounts of time (Fig. 7A, B). At low passage, almost all chromosomes had a normal structure (Fig. 7B and Supplementary Fig. 7A), but after approximately 50 passages, the pericentromeric region upstream of the neocentromere had undergone break and or fusion events in 70% of cells. Copy number analysis was used to estimate the position of breakage events (Supplementary Fig. 7B) which were predominantly in the vicinity of the centromere. However, a region of pronounced DNA amplification was visible near the *DIPK2A* gene located proximal to the pericentromere boundary (Supplementary Fig. 7C). FISH of the cell population with the border fosmid indicated this region was amplified and located on other chromosomes but strikingly was rarely visible (2%) after chromosome breakage and fusion events (Supplementary Fig. 6D) indicating that breaks occurred in the pericentromeric chromatin domain upstream of the neocentromere.

### Discussion

For efficient and accurate chromosome segregation kinetochores must assemble onto CENP-A-containing chromatin. This happens in two stages, initially, the constitutive centromere-associated network (CCAN)[71] binds to centromeric chromatin via CENP-C[28]. Then, in mitotsis, the complete kinetochrore is assembled to provide an attachment site for the microtubules. To achieve these two steps with high fidelity, it has been speculated that the underlying chromatin must adopt a special or distinct chromatin structure[2] (Fig. 7C).

Previous studies have indicated that centromeric chromatin is associated with histone marks that are reflective of actively transcribed chromatin[13,72]. After centromere repositioning, we observe a strong enrichment of active histone marks (Fig. 1) and a significant recruitment of RNA polymerase (Fig. 2). Concomitantly, the chromatin fibre is remodelled to form a disrupted or 'open' structure (Fig. 4A); but what are the mechanisms for forming disrupted chromatin and what role might it play in kinetochore formation and function? Gene-rich[59] and transcriptionally active[60] chromatin are reported to form disrupted chromatin fibre structures, through a combination of mechanisms. At typical euchromatic regions irregularly positioned nucleosomes are less able to fold into an organised chromatin structure, but at

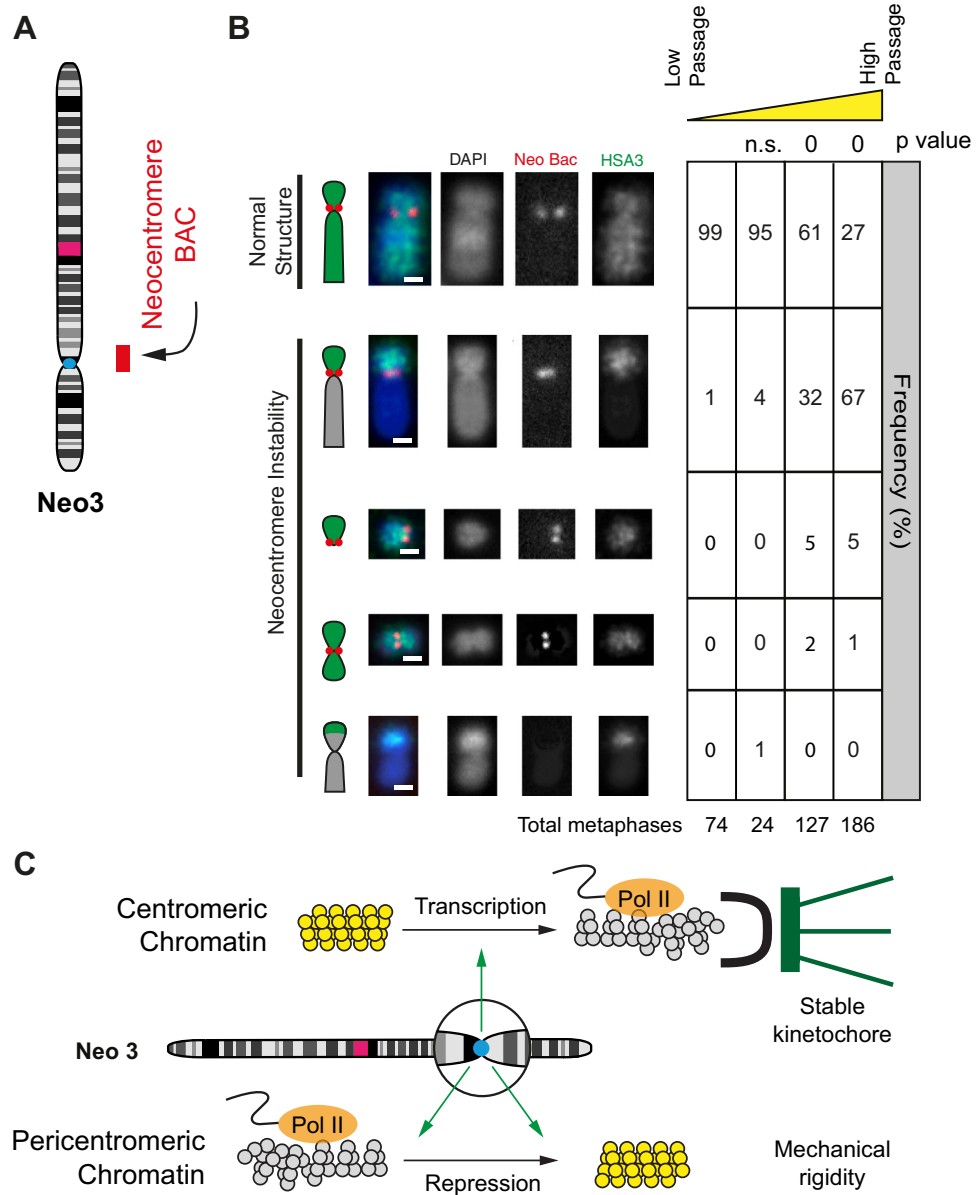

**Fig. 7 | Neocentromere chromosome instability. A** Neo3 ideogram and BAC probe used to analyse chromosome stability. **B** Left, representative 2D DNA FISH images of Neo3 metaphase chromosomes hybridised to a human chromosome 3 paint (green) and BAC (red) located at the neocentromere. Chromosome morphology was scored as normal or showing instability: deletions, fusions or duplications. The bar is 2 μm. Right, quantification (%) of different chromosome morphologies with increasing passage number (low -10, high >50) over time. *P* values are for a $\chi^2$ test compared to low passage. **C** Model showing transcription-dependent centromere remodelling to a disrupted euchromatin state (grey) and pericentromeric chromatin repression to form heterochromatin (yellow). We suggest that disrupted euchromatin provides a suitable foundation for a high-fidelity kinetochore whilst heterochromatin and the accumulation of satellite sequences generate surrounding mechanical rigidity.

centromeric regions it is known that satellite DNA sequences position nucleosomes regularly and form a rigid chromatin fibre[8,11]. After centromere repositioning there was no apparent remodelling of nucleosomes (Fig. 5A), suggesting that this is not the basis for the disrupted chromatin fibre. Alternatively, it is reported that CENP-A nucleosome tails bind DNA less tightly to form more dynamic nucleosomal structures and may also interfere with linker histone binding, to promote chromatin fibre opening[73]. Although CENP-A nucleosome properties might influence chromatin fibre folding, it appears that the disrupted chromatin structure is strongly transcription dependent (Fig. 4B). We, therefore, speculate that transcription could disrupt nucleosome positioning through the activity of RNA polymerase and recruitment of chromatin remodelling machines.

At centromeres, disrupted chromatin fibres may serve two purposes. Firstly, a disrupted structure might increase the likelihood of proteinaceous components of the CCAN, such as CENP-C, binding to CENP-A-containing centromeric nucleosomes[28] but may also facilitate other structural components such as RNA to interact due to increased access to histone proteins. Our data indicate that centromere chromatin structure is RNA dependent demonstrating that additional nucleic acids may play a structural role (Fig. 6). This is consistent with previous studies, which show that RNA can interact with HP1 to facilitate heterochromatin folding[67]. Secondly, depending on the nature of centromeric chromatin, a flexible fibre might be able to adopt a structure that is able to form a better scaffold for kinetochore formation. For example, in one model it has been suggested that

centromeric chromatin might form a layered configuration termed a boustrophedon[74] whilst in an alternate model, the centromeric chromatin might be folded into small loops[14], creating an invaginated structure that CCAN proteins can securely attach to. Presumably, both large-scale structures would form more readily from a flexible chromatin fibre.

Another recent idea posits that heterochromatin can undergo liquid-liquid phase separation (LLPS)[75,76] to form a gel-like microenvironment[77] that could facilitate kinetochore assembly. LLPS often occurs through non-covalent interactions that can be modulated by the local concentration of RNA and proteinaceous components such as HP1 or the long tails of histones[78,79], so might be facilitated by a disrupted underlying chromatin structure.

By analysing a single neocentromere (Neo3), this study indicates that the centromeric chromatin core has a flexible disrupted structure (Figs. 1 and 4) flanked by transcriptionally repressed pericentromeric chromatin (Figs. 1 and 5), to form a two-domain model (Fig. 7C). Although chromatin fibre decompaction was also observed at a second neocentromere (Neo6) (Supplementary Fig. 6) lack of available human-hamster hybrid cell lines prohibited further analysis of this neocentromere, therefore, it is not possible to rule out that some of our findings cannot be generalised beyond Neo3. At a newly formed (neo) centromere these flanking regions are epigenetically remodelled and transcriptionally repressed (Fig. 1), presumably by an activity emanating from the core, but epigenetic remodelling was insufficieint to completely compact chromatin fibre structure (Fig. 5), as observed for satellite-containing heterochromatin[8]. Evidence from evolutionary new centromeres (ENCs) indicates that satellite sequences accumulate over a long period[40–42]. Consistently only a small region of pericentromeric chromatin had a compact structure (Fig. 5), showing that the neocentromere at 3q24 is young and has not yet matured to adopt a compact structure. Concomitantly, it exhibited a low level of aneuploidy, suggesting that ENCs recruit satellite DNA sequences over time to progressively form a stable chromatin platform[8] (Fig. 7C). We therefore suggest that centromere repositioning is accompanied by significant transcriptional, epigenetic and chromatin fibre remodelling to form a suitable environment for kinetochore assembly and that over time the chromatin fibre structure matures to support high fidelity chromosome segregation in mitosis[66,80,81].

## Methods

### Cell lines
The human parental lymphoblastoid cell line was grown in RPMI 1640 with L-Glutamine (Life Technologies) supplemented with 20% FCS, penicillin (100 U.ml$^{-1}$) and streptomycin (100 µg.ml$^{-1}$). Human/Hamster hybrid cell lines GM10253A and HybNeo3, harbouring HSA3 and Neo3, respectively, were grown in the same media but with 10% FCS. All cells were maintained at 37 °C in an atmosphere of 5% CO2 and subjected to regular mycoplasma testing. Transcription was blocked by adding α-amanitin (50 µg.ml$^{-1}$) or Flavopiridol (100 µM) to cells for the times indicated.

### DNA fluorescence in situ hybridisation (FISH)
FISH was performed on both metaphase chromosome spreads and interphase nuclei. Metaphase chromosomes were prepared by treating cells with 0.1 µg.ml$^{-1}$ Colcemid (Life Technologies, Cat No 15210-040) for 30 min (hybrid cells) or 4 hr (parental lymphoblastoid cells) prior to harvest to induce mitotic arrest and increase the number of mitotic cells. Cells were recovered by trypsin treatment and washed in PBS. Hypotonic solution, containing 75 mM KCl was added dropwise to a final 5 ml volume. Hypotonic treatment was performed at room temperature for 10 min, after which cells were pelleted by centrifugation at 1200 rpm (200 g) for 5 min and fixed three times in 5 ml of a freshly prepared solution of 3:1 ratio (v/v) methanol: acetic acid (MAA). The MAA fixative was added to the cell pellet dropwise with constant

agitation. Chromosome preparations were stored at −20 °C. To prepare slides with metaphase spreads, metaphase chromosome preparations were dropped onto glass slides. The glass slides were pretreated in a dilute solution of HCl in Ethanol for at least one hour prior to use. The chromosome preparations were pelleted by centrifugation at 1200 rpm (200 g) for 5 min and resuspended in freshly prepared MAA solution until the suspension became cloudy. Two drops of the suspension were dropped onto a pre-treated glass slide from a height of 20 cm and dried at room temperature overnight before staining or hybridisation.

For 3D FISH on interphase nuclei hybrid cells were grown overnight on glass slides whilst parental non-adherent lymphoblastoid cells ($3 \times 10^4$ cells) were cytospun onto glass slides at 600 rpm (50 g) for 10 min. Slides were rinsed with PBS and fixed in 4% paraformaldehyde (PFA) for 10 min. Slides were then rinsed with PBS and cells were permeabilized for 10 min on ice with PBS supplemented with 0.2% triton. After rinsing, slides were stored in PBS (for immunohistochemistry) or 70% ethanol (for FISH) at 4 °C. For chromatin nicking and RNase H treatment cells were grown on slides overnight, rinsed gently whilst still in the slide tray three times with PBS and then treated with bleomycin (100 µM) in PBS or RNase H (1000 U.ml$^{-1}$, NEB MO297S) in PBS supplemented with 0.1% triton, 1 mM Ca$^{2+}$ and 1 mM Mg$^{2+}$ for 10 min at 37 °C. Slides were then rinsed with PBS and PFA fixed as before.

FISH was carried out as described[65] except that MAA fixed metaphase spreads were denatured for 1 min in 70% formamide in 2x SSC, pH 7.5, at 70 °C, and interphase cells (grown on glass slides or cytospun) were 4% PFA fixed and were denatured for 45 min at 80 °C. Following denaturation, slides were submerged in ice-cold 70% ethanol for 2 min and then dehydrated through 90 and 100% ethanol for 2 min each at room temperature. Fosmid and BAC clones (BACPAC Genomics) were labelled by nick translation with digoxigenin-11-dUTP (Roche, #11093070910) or biotin-16-dUTP (Roche, #11093088910) for antibody-based detection as previously described or alternatively directly labelled with Green 500 dUTP (Enzo-42845) or red-dUTP (ChromaTide Alexa Fluor 594-5-dUTP C11400). α-satellite probe p82H[82] was labelled by nick translation. For hybridisation, 150 ng of labelled probe was combined with 5 µg salmon sperm and 10 µg human C$_0$t1 DNA (Invitrogen, Cat No 15279011). Two volumes of ethanol were added and the probe mix was collected by centrifugation and dried. Dried probes were resuspended in 10 µl of hybridisation buffer containing 50% formamide (v/v), 1% Tween-20 and 10% dextran sulphate (Sigma Aldrich, Cat No D8906-100G) in 2x SSC. Chromosome 3 paint (XCP3 Green, Metasystems) was supplied already labelled with a green fluorophore and dissolved in hybridisation solution and ready to use. Probes were denatured at 70 °C for 5 min and reannealed at 37 °C for 15 min and chilled on ice. Probes were pipetted onto slides and hybridisation was performed at 37 °C overnight. Coverslips were removed and slides were washed four times in 2x SSC at 45 °C for 3 min and four times in 0.1x SSC at 60 °C for 3 min. Slides were blocked in 5% milk in 4x SSC for 5 min at RT. Detection of biotin label was performed with sequential layers of fluorescein (FITC)-conjugated avidin (Vector, A-2011; 1:500), biotinylated anti-avidin (Vector, BA-0300; 1:100) and a further layer of FITC-avidin (Vector, A-2011; 1:500). Digoxigenin was detected with sequential layers of Rhodamine-conjugated anti-digoxigenin (Roche, 11 207 750 910; 1:20) and Texas-Red (TR) –conjugated anti-sheep IgG (Vector, TI 6000; 1:100). Slides were DAPI stained, mounted in Vectashield (Vector Laboratories, Cat No H-1000) Epifluorescent images were acquired using a Photometrics Coolsnap HQ2 CCD camera on a Zeiss Axioplan II fluorescence microscope with a Plan-neofluar/apochromat 100x objective (Carl Zeiss, Cambridge, UK), a Mercury Halide fluorescent light source (Exfo Excite 120, Excelitas Technologies) and Chroma #83000 triple band pass filter set (Chroma Technology Corp., Rockingham, VT) with the single excitation and emission filters installed in motorised filter wheels (Prior Scientific

Instruments, Cambridge, UK). Data was collected using Micromanager software and analyzed using custom scripts in iVision (Version 4.5.6 r4).

For four-colour 3D Immuno-FISH immunocytochemistry for CENP-C was performed prior to FISH. Slides stored in PBS were blocked in 5% horse serum then incubated overnight with anti-CENP-C antibody[83] (non-commercial antibody provided by Stefania Purgato; 1:200) before 1 hr incubation with Texas Red labelled anti-rabbit (1:100, Jackson ImmunoResearch Laboratories) secondary antibody. CENP-C signal was fixed with 4% PFA for 45 min followed by denaturation with 50% formamide in 2x SSC, pH 7.5, at 80 °C for 45 min. Slides were then dipped briefly in 2x SSC followed by incubation overnight at 37 °C with pairs of labelled fosmid probes. Slides were then washed and processed as above. Epifluorescent images were acquired using a Photometrics Coolsnap HQ2 CCD camera and a Zeiss AxioImager A1 fluorescence microscope with a Plan Apochromat 100×1.4NA objective, a Nikon Intensilight Mercury-based light source (Nikon UK Ltd, Kingston-on- Thames, UK) and a Chroma 89000ET single excitation and emission filters (Chroma Technology Corp., Rockingham, VT) with the excitation and emission filters installed in Sutter motorised filter wheels (Sutter Instrument, Novato, CA). A piezoelectrically driven objective mount (PIFOC model P-721, Physik Instruments GmbH & Co, Karlsruhe) was used to control movement in the z dimension. Hardware control, image capture and analysis were performed using Nikon Nis-Elements software (Nikon UK Ltd, Kingston-on-Thames, UK) and Volocity (Perkinelmer, Inc.). Images were deconvolved using a calculated point spread function with the constrained iterative algorithm of Volocity. Image analysis was carried out using Imaris software that calculates the distance between two fosmid probe signals.

The significance of compaction between pairs of probes was tested using the nonparametric Wilcoxon test for paired samples, $P < 0.05$ was considered significant.

FISH probes are described in Supplementary Table 1.

## Immunocytochemistry

Metaphase chromosome spreads derived from parental cells were rinsed in PBS, blocked in 5% horse serum then incubated overnight with anti-CENP-C antibody[83] (non-commercial antibody provided by Stefania Purgato; 1:200) and anti-RNA pol II (1:1000, Abcam Ab24758) antibody. Secondary antibodies were FITC-conjugated anti-mouse and Texas Red-conjugated anti-rabbit antibodies (1:150, Jackson ImmunoResearch Laboratories). Slides were DAPI stained, mounted in Vectashield (Vector Laboratories, Cat No H-1000) and imaged on a Zeiss epifluorescence microscope using a 100x objective.

## Chromatin immunoprecipitation

ChIP was done as described[84] except that a Soniprep 150 (Sanyo) was used for sonication. In brief, cells ($5-6 \times 10^6$ in 10 cm dishes) were cross-linked with 10 ml 1% formaldehyde (Sigma) in medium for 5 min at room temperature and then incubated in 10 ml 200 mM glycine in the medium for 5 min. Cells were rinsed twice with PBS and incubated with 7 ml lysis buffer (10 mM Tris-HCl (pH 7.5), 10 mM NaCl and 0.5% NP-40) for 10 min at room temperature with mild rotation. This lysis buffer was then aspirated off, and cells were scraped into 1 ml lysis buffer and centrifuged at 3000 rpm (845 g) for 3 min at 4 °C. Cell pellet was resuspended in 100 µl SDS lysis buffer (50 mM Tris-HCl (pH 7.5), 10 mM EDTA and 1% SDS) and mixed by pipetting. 400 µl ChIP dilution buffer (50 mM Tris-HCl (pH 7.5), 167 mM NaCl, 1.1% Triton X-100, 0.11% sodium deoxycholate and protease inhibitor cocktail (complete EDTA-free; Roche)) was added before sonication (fifteen times for 20 s at 2 µm). After centrifugation at 13,000 rpm (15871 g) for 15 min at 4 °C to remove the insoluble material, the supernatant was removed to a new 1.5 ml tube and the volume made up to 500 µl with ChIP dilution buffer. 50 µl was removed as input for ChIP and the rest of the sample was incubated with antibody-bound Dynabeads overnight at 4 °C with rotation. Dynabeads were prepared in advance by taking 50 µl of beads and washing three times with 500 µl cold RIPA-150 mM NaCl buffer (50 mM Tris-HCl (pH 7.5), 150 mM NaCl, 1 mM EDTA, 1% Triton X-100, 0.1% SDS, 0.11% sodium deoxycholate and protease inhibitor cocktail). Beads were then incubated with 500 µl cold RIPA-150 mM NaCl buffer plus antibody for 2 hr at 4 °C with rotation. Beads were then washed three times with 500 µl cold RIPA-150 mM NaCl buffer and were then ready for overnight incubation with the ChIP sample. Beads were washed sequentially with 1 ml cold RIPA- 500 mM NaCl (50 mM Tris-HCl (pH 7.5), 500 mM NaCl, 1 mM EDTA, 1% Triton X-100, 0.1% SDS, 0.11% sodium deoxycholate and protease inhibitor cocktail) and twice with 1 ml TE (10 mM Tris-HCl (pH 8.0) and 1 mM EDTA). DNA was eluted by the addition of 200 µl ChIP direct elution buffer (10 mM Tris-HCl (pH8.0), 300 mM NaCl, 5 mM EDTA and 0.5% SDS) and incubated overnight at 65 °C. Samples were then treated with DNase-free RNase (Roche; 5 µg.ml$^{-1}$; 37 °C; 30 min) and proteinase K (250 µg.ml$^{-1}$; 55 °C; 1 h). DNA was extracted with phenol-chloroform-isoamyl alcohol (25:24:1) and ethanol precipitated with carrier (1 µl glycogen, Invitrogen) on dry ice for 30 min. Following 70% ethanol wash the DNA pellet was resuspended in 20 µl water and quantified using a NanoDrop. For microarray hybridisation, immunoprecipitated DNA was amplified using whole-genome amplification (Sigma).

Magnetic sheep anti-mouse IgG beads (Dynabeads, Invitrogen, 11201D) were used for mouse antibodies and protein G beads were used for rabbit antibodies (Dynabeads, Invitrogen, 10004D). Antibodies used were to CENP-A[83] (non-commercial antibody provided by Stefania Purgato; 1:100) and CENP-C[83] (non-commercial antibody provided by Stefania Purgato; 1:100), H3K27ac (Abcam, Ab4729; 2 µg for 25 µg of chromatin), H3K4me2 (Millipore, 07−030; 5 µl for 25 µg of chromatin), H3K9me2 (Millipore, 07−212; 4 µg for 25 µg of chromatin), H3K9me3 (Abcam, Ab8898; 4 µg for 25 µg of chromatin), H3K27me3 (Abcam, Ab6002; 5 µg for 25 µg of chromatin), H4K20me1 (Abcam, Ab9051; 2 µg for 25 µg of chromatin), H3K36me3 (Abcam, Ab9050; 4 µg for 25 µg of chromatin), CTCF (Cell Signalling Technology, 2899; 4 µl for 25 µg of chromatin), RNA Polymerase II (Diagenode, C15200004; 1 µg for 25 µg of chromatin), RNA Polymerase II (gift from H. Kimura; 5 µg for 25 µg of chromatin). All antibodies were characterised using western blots, and ChIP was optimised using quantitative PCR assays.

## Analysing changes in DNA supercoiling

Biotinylated psoralen (bTMP) uptake was used to analyse DNA supercoiling as previous described[65]. Cells were treated with 500 µg.ml$^{-1}$ of bTMP in PBS for 20 min at room temperature in the dark. bTMP was UV cross-linked to DNA at 360 nm for 10 min. DNA was purified from cells using SDS and proteinase K digestion and extracted using phenol-chloroform-isoamyl alcohol (25:24:1). DNA was fragmented by sonication (thirteen times for 30 s at 2 µm). Biotin incorporation into DNA was detected by dot blotting using alkaline phosphatase−conjugated avidin as a probe. The bTMP−DNA complex in TE was immunoprecipitated using avidin conjugated to magnetic beads (Dynabeads MyOne Streptavidin Invitrogen, 65001) for 2 h at room temperature and then overnight at 4 °C. Beads were washed sequentially for 5 min each at room temperature with TSE I (20 mM Tris-HCl, pH 8.1, 2 mM EDTA, 150 mM NaCl, 1% Triton X-100 and 0.1% SDS), TSE II (20 mM Tris-HCl, pH 8.1, 2 mM EDTA, 500 mM NaCl, 1% Triton X-100 and 0.1% SDS) and buffer III (10 mM Tris-HCl, pH 8.1, 0.25 M LiCl, 1 mM EDTA, 1% NP40 and 1% deoxycholate). Beads were then washed twice with TE buffer for 5 min. To extract DNA and to release psoralen adducts, the samples were boiled for 10 min at 90 °C in 50 µl of 95% formamide with 10 mM EDTA. Samples were then made up to 200 µl with water, and the DNA was purified using a Qiagen MinElute PCR purification kit. bTMP bound DNA was amplified using whole-genome amplification (Sigma) prior to microarray hybridisation.

## Chromatin fractionation

Disrupted or 'open' chromatin was isolated as described previously[60]. In brief, cell nuclei were digested with micrococcal nuclease and soluble chromatin released overnight followed by fractionation on a 6–40% isokinetic sucrose gradient in 80 mM NaCl, 0.1 mM EDTA, 0.1 mM EGTA and 250 μM PMSF. DNA purified from gradient fractions was analyzed by electrophoresis through 0.7% agarose in 1x TPE buffer (90 mM Tris-phosphate, 2 mM EDTA) with buffer circulation. Preparative fractionation of DNA from gradient fractions was carried out by pulsed-field gel electrophoresis (PFGE) (CHEF system, Biorad) through 1% low melting point agarose in 0.5 × TBE, at 180 V, for 40 h, with a 0.1–2 s switching time. Size markers were 1 kb (Promega) and λ HindIII (NEB) DNA ladders. EtBr-stained gels were scanned using a 473 nm laser and a 580 nm band-pass filter on a Fuji FLA-3000. DNA of ~20 kb, corresponding to "open" chromatin, was isolated by β-agarase (NEB) digestion and amplified by whole genome amplification (Sigma) prior to microarray hybridisation.

## Microarray hybridisation, data processing and analysis

Whole-genome amplified DNA (ChIP/bTMP/'open' chromatin) was labelled and hybridised as previously[65] to custom 180 K Agilent microarrays (7 Mb spanning the neocentromere domain (chr3:142781158–149782213; GRCh38 (hg38). In brief, 500 ng DNA was random prime labelled (ENZO) with Cy3 (Sample DNA) or Cy5 (Input DNA) and purified on a MinElute PCR purification column (Qiagen). Labelled DNA was diluted in hybridisation buffer (Agilent) and hybridised to arrays for 24 h at 65 °C. Slides were washed according to the manufacturer's instructions and scanned on a Nimblegen Microarray scanner at 2 μm resolution generating a TIFF file.

Spot signal intensity was extracted from the TIFF files using Agilent Feature Extraction software and were pre-processed in R using the RINGO bioconductor package to give the raw Cy5 and Cy3 signal intensities for each spot. Individual Cy5 and Cy3 channels were normalised to each other and between arrays using a variance stabilising algorithm (for bTMP arrays) and loess, vsn (for ChIP arrays) or nimblegen ("open" chromatin arrays) normalised and scaled, using the standard Bioconductor LIMMA package. All arrays were quality controlled by checking array hybridisation patterns, analysing signal profiles and using MA plots. For data analysis log2(sample/input) data was loaded in to the ZOO package in R and for display the data was smoothed using a rolling median.

## RNA extraction and RNA-Seq

Total RNA was extracted from cells using RNeasy mini kit (Qiagen) with on-column DNase I digestion (RNAse-Free DNase Set, Qiagen). For RT-qPCR RNAs were reverse transcribed (Superscript II, Invitrogen) using random primers and quantified by qPCR (Fast start SYBR green, Roche). Primer sequences are described below. For RNA-seq Qiagen miRNeasy kit was used to separately extract short (miRNA enriched) and long RNA (total RNA > 200nt). RNA was sized and quality controlled on an RNA ScreenTape (Agilent). Ribosomal RNA was depleted using Illumina® Ribo-Zero Plus rRNA Depletion Kit (Illumina, 20040526) following the manufacturer's instructions, and libraries for RNA-seq were prepared and indexed using NEBNext® Ultra™ II DNA Library Prep Kit for Illumina® (NEB #E7645L) and NEBNext Singleplex Oligos for Illumina (NEB #E7335, E7500) following the manufacturer's instructions. Libraries were sized and quality controlled on a D1000 Tapestation tape (Agilent). Single-end RNA-seq of 50 bp read length was performed on Illumina Hi-Seq 2000 (UMC, Amsterdam). FASTQ sequence files were obtained and the RNA-seq reads were aligned to the human reference genome (hg38) using TopHat v2 and Bowtie2. Aligned BAM files were processed with Samtools v1.6[85] and the Bedtools "genome coverage" tool[86]. Sequence read depth for short RNAs were 28 million for each of replicate (two replicates) with mapping

efficiencies of approximately 30%. Sequence read depth for long RNAs were 12 million for each of the replicate (two replicates) with mapping efficiencies of >80%.

## TT-seq

Nascent RNA was labelled by adding 500 μM 4-thiouridine (4sU) (Sigma, T4509) to cells harbouring HSA3 and Neo3 in T75 flasks and incubating at 37 °C for 10 min. Media was aspirated and RNA extraction was performed with TRIzol (Invitrogen) following the manufacturers' instructions. After DNase treatment (Turbo DNase, Thermo Fisher Scientific) RNA concentration and purity were determined using a NanoDrop. RNA (70 μg) was fragmented in 100 μl $H_2O$ to <1.5 kb by 20 cycles of 30 s on/30 s off at high power in a Biorupter plus and RNA size assessed by agarose gel electrophoresis. Fragmented 4sU labelled RNA was biotinylated by adding 140 μl of EZ-Link Biotin-HPDP (1 mg.ml$^{-1}$ in dimethylformamide; Pierce, 21341), 70 μl of 10x biotinylation buffer (100 mM Tris-HCl pH 7.5, 10 mM EDTA) and $H_2O$ to a final volume of 700 μl. This was incubated at room temperature for 1.5 h with rotation. Unincorporated biotin-HPDP was removed by two rounds of chloroform extraction with 2 ml Phase lock gel heavy tubes (Eppendorf). RNA was precipitated with 1/10 volume of 5 M NaCl and an equal volume of Isopropanol. This was inverted to mix and incubated at room temperature for 10 min followed by centrifugation at 10,000 g for 20 min at room temperature. RNA pellet was washed with 80% EtOH and centrifuged at 13,000 rpm (15871 g) for 10 min at 4 °C. RNA was resuspended in 100 μl $H_2O$ and dissolved by heating to 40 °C for 10 min with agitation. RNA was then immediately placed on ice and RNA concentration determined by Nanodrop spectrophotometer. Biotinylated 4sU labelled RNAs were then recovered using μMACS Streptavidin MicroBeads (Miltenyi, 130-074-101) and separation on a μMACS Separator. For the concentration of total RNA in μg per sample an equal amount in μl of Streptavidin microbeads was added. This was incubated at room temperature for 15 min with rotation. μMacs columns were equilibrated with 900 μl room temperature washing buffer (100 mM Tris-HCl pH 7.5, 10 mM EDTA, 1 M NaCl, 0.1% Tween20). The RNA/streptavidin bead solution was then applied to the column followed by three washes with 900 μl of washing buffer at 65 °C and three washes with 900 μl of washing buffer at room temperature. RNA was eluted with 2 × 100 μl of fresh elution buffer (100 mM dithiothreitol in RNase-free $H_2O$) directly into 2 ml lobind tubes (Eppendorf) containing 700 μl Buffer RLT (RNeasy MinElute Cleanup Kit, Qiagen). Five hundred microlitres of 100% ethanol was added to the RNA solution, and mixed thoroughly by pipetting before RNA was purified through RNAeasy MinElute Spin Columns. RNA concentration was determined using a Nanodrop and libraries for RNA-seq were prepared and indexed using NEBNext® Ultra™ II Directional RNA Library Prep Kit for Illumina® (NEB #E7645L) and NEBNext Singleplex Oligos for Illumina (NEB #E7335, E7500) following the manufacturer's instructions. Libraries were sized and quality controlled on a D1000 Tapestation tape (Agilent) and Illumina sequencing (paired-end RNA-seq of 50-bp read length) was performed on NovaSeq S1 (Edinburgh Genomics). FASTQ sequence files were obtained and the RNA-seq reads were aligned to a Human(hg38) /Hamster (GCA_003668045.1) hybrid reference genome using Bowtie2 and processed with Samtools v1.6[85], and the deepTools "bamCoverage" tool[86] with RPKM normalisation. Sequence read depth for HSA3 control replicates was 74 million and 78 million with mapping efficiencies of 56 and 64%. For HSA3 plus exosome knockdown the read dept per replicate was 72 million and 64 million with 65% and 60.6% mapping efficiency. Sequence read depth for Neo3 control replicates was 45 million and 93 million with mapping efficiencies of 60 and 70%. For Neo3

plus exosome knockdown the read dept per replicate was 38 million and 128 million with 65 and 63% mapping efficiency.

## Exosome RNA interference

For siRNA treatment, cells (GM10253A and HybNeo3; 10–20% confluent) were transfected with 10 nM Silencer Select Pre-designed siRNA targeting EXOSC3 (Ambion, Life Technologies) using Lipofectamine RNAi MAX (ThermoFisher) 24 h after seeding and again 48 h later. After a further 48 h exosome knockdown was confirmed by western blotting and TTseq was performed. Silencer Select RNA sequence for EXOSC3 were GAGATATATTCAAAGTTGA, part number s83102. The control RNA was Stealth RNAi siRNA Negative Control (ThermoFisher). For western blotting cells were suspended in NuPAGE LDS sample buffer (ThermoFisher) with 10 mM DTT, incubated at 100 °C for 5 min and sonicated briefly. Protein samples were resolved on 12% bis-tris gels (ThermoFisher) and transferred to Immobilon-P PVDF 0.45 mm membrane (Merck Millipore) by wet transfer. Membranes were probed with anti-EXOSC3 antibody (Abcam, Ab156683; 1:1000) using standard techniques and detected by enhanced chemiluminescence.

## Shallow DNA sequencing

The hybrid HSA3 and Neo3 cell lines were shallow sequenced to confirm copy number. Genomic DNA was prepared from cells and 500 ng DNA was fragmented using a Covaris sonicator. Genomic DNA libraries were prepared using Illumina TruSeq Nano DNA LT sample prep kit as per the manufacturer's instructions and Illumina sequencing (50 bp, single-end reads) was performed on Illumina Hiseq 4000 (VUMC Cancer Centre, Amsterdam). FASTQ sequence files were obtained and reads were aligned to the human reference genome (hg19) using BWA and processed with Samtools v1.6[85]. In R the BAM files were loaded into the Bioconductor package QDNAseq for copy number analysis. Human reference genome HG19 was used here as QDNAseq has pre-calculated bin annotations for genome build hg19. Sequence coverage was 0.23x coverage/base for HSA3 and 0.32x coverage/base for Neo3.

## Neocentromere capture DNA sequencing

NimbleGen Sequence Capture technologies were employed for targeted deep sequencing of the neocentromere domain. Capture probes tiling a 1.5 Mb domain across the neocentromere were designed using Nimblegen capture design software, and sequence capture was performed using this custom SeqCap EZ Choice probe pool and SeqCap EZ HE-Oligo Kit A and SeqCap EZ Accessory Kit (Nimblegen) according to manufacturer's instructions. In brief, genomic DNA (gDNA) from the parental lymphoblastoid cell line was fragmented to ~200–500 bp with 20 cycles of 30 s on/30 s off on a Biorupter. 1 µg gDNA was then used to prepare the gDNA sample library using NEBNext® Ultra™ II DNA Library Prep Kit for Illumina® (NEB #E7645L), following the manufacturer's instructions. The neocentromere domain was captured by hybridising this gDNA library with the biotinylated SeqCap EZ library. 342 ng of gDNA library was mixed with 5 µg $C_0t1$ DNA, 1000 pmol of SeqCap HE Universal Oligo 1 and 1000 pmol SeqCap HE Index Oligo, 7.5 µl 2 X Hybridisation Buffer and 3 µl Hybridisation Component A. This was vortexed for 10 s and centrifuged at maximum speed for 10 s before denaturing at 95 °C for 10 min. This gDNA/$C_0t1$/Oligo/Hybridisation cocktail was then combined with the SeqCap EZ library (provided as 4.5 µl single-use aliquots in 0.2 ml tubes), vortexed for 3 s and centrifuged at maximum speed for 10 s. Hybridisation was then performed on a thermocycler at 47 °C and incubated for 70 h. Each hybridisation reaction was then bound to streptavidin beads from SeqCap EZ Pure Capture Bead Kit and washed with SeqCap EZ Hybridisation and Wash Kit (Nimblegen), following the manufacture's protocol. Captured libraries were re-amplified using Post LM-PCR oligos (Nimblegen) and Q5 High-Fidelity DNA polymerase (NEB) directly from the beads. A mastermix consisting of 65 µl 2x NEBNext Ultra II Q5 Master Mix NEB, 50 µl captured library (beads in H₂0), 13 µl LM-PCR Oligo mix (Oligo 1 &

2, 2 µM final concentration of each)(Nimblegen) was made up to 50 µl with H₂O, vortexed to mix and then split into 2 × 65 µl samples for PCR using the following PCR cycling conditions. Initial incubation at 98 °C for 30 s, 14 cycles of 98 °C for 10 s, 65 °C for 30 s and 72 °C for 30 s. Final incubation of 72 °C for 5 min and hold at 4 °C. The two PCR reactions were recombined and the captured DNA was purified using 1.8:1 AMPure XP Beads: DNA ratio. Capture efficiency was determined to be between 94 and 117 fold using a Nimblegen Sequence Capture control locus qPCR assay. Neocentromere captured DNA libraries were sized and quality controlled on a D1000 Tapestation tape (Agilent), and paired-end sequenced (50 bp) on an Illumina MiSeq (Edinburgh Genomics).

## Nucleosome positioning

Nuclei were extracted from cells carrying HSA3 and Neo3 as described[8] and resuspended in NB-R (85 mM KCl, 10 mM Tris-HCl [pH 7.6], 5.5% (w/v) sucrose, 1.5 mM CaCl₂, 3 mM MgCl₂, 250 µM PMSF). Nuclei (800 µl at 5A260) were digested with DFF nuclease (PMID: 17626049) for increasing amounts of time (100 µl of digested nuclei were removed to a new tube after 1, 2, 4, 8, 16 and 32 min digestion) at room temperature in the presence of 100 µg.ml⁻¹ RNaseA. Digestion was stopped by adding EDTA to 10 mM. DNA was purified with SDS/Proteinase K digestion, phenol/chloroform extraction and ethanol precipitation. Agarose gel electrophoresis of DFF digested nuclei confirmed digestion to mono and di-nucleosomes. The 4 min and 8 min samples and the 16 min and 32 min samples were pooled and ran on 2.5% LMP GTG agarose in 1× Sybr Safe (Thermo Fisher) dye. Mono and di-nucleosome DNA bands were excised from the gel and purified by β-agarase (NEB) digestion followed by phenol/chloroform extraction and ethanol precipitation. 500 ng DNA samples (50 ng from the 4 and 8 min DFF digestion pool plus 450 ng of the 16 and 32 min DFF digestion pool) were concentrated to 55 µl volume using 2:1 AMPure XP Beads: DNA ratio for the mono nucleosome samples and 1.6:1 AMPure XP Beads: DNA ratio for the di nucleosome samples. Genomic DNA sample libraries were then prepared using NEBNext® Ultra™ II DNA Library Prep Kit for Illumina® (NEB #E7645L), following the manufacturer's instructions. This included adaptor-ligated DNA size selection of between 100–200 bp for the mono nucleosome samples and 300–400 bp for the di nucleosome samples. DNA library yield was increased by a further round of PCR with Post LM-PCR oligos (Nimblegen) and Q5 High-Fidelity DNA polymerase (NEB). The PCR reaction consisted of 30 µl of the mono or di nucleosomal DNA libraries (150–270 ng), 50 µl 2× NEBNext Ultra II Q5 Master Mix NEB, 5 µl Post LM-PCR oligo mix (Oligo 1 & 2, 2 µM final concentration of each; Nimblegen) and 15 µl H₂O. PCR cycling conditions were an initial incubation at 98 °C for 30 s, 5 cycles of 98 °C for 10 s and 65 °C for 75 s and a final incubation of 65 °C for 5 min. DNA was purified using 1.8:1 AMPure XP Beads: DNA ratio and libraries were quantified and sized on a D1000 Tapestation tape (Agilent). Libraries, representative of mono and di nucleosome positions throughout the genome, were pooled in equimolar amounts (150 nM) and then subjected to neocentromere capture (as above) to examine nucleosome positioning across the neocentromere region. 1.25 µg of the mono and di nucleosome library pool was mixed with 5 µg $C_0t1$ DNA, 1000 pmol of SeqCap HE Universal Oligo 1 and 1000 pmol SeqCap HE Index Oligos 14, 16, 18 and 19, 7.5 µl 2 × Hybridisation Buffer and 3 µl Hybridisation Component A. This was vortexed for 10 s and centrifuged at maximum speed for 10 s before denaturing at 95 °C for 10 min. This gDNA/$C_0t1$/Oligo/Hybridisation cocktail was then combined with the SeqCap EZ library (provided as 4.5 µl single-use aliquots in 0.2 ml tubes), vortexed for 3 s and centrifuged at maximum speed for 10 s. Hybridisation was then performed on a thermocycler at 47 °C and incubated for 70 h. Each hybridisation reaction was then bound to streptavidin beads from SeqCap EZ Pure Capture Bead Kit and washed with SeqCap EZ Hybridisation and Wash Kit (Nimblegen), following the manufacturer's protocol. Captured

libraries were re-amplified using Post LM-PCR oligos (Nimblegen) and KAPA High-Fidelity DNA polymerase (KAPA Biosystems) directly from the beads. 25 μl 2x KAPA HiFi HotStart ReadyMix, and 5 μl LM-PCR Oligo mix (Oligo 1 & 2, 2 μM final concentration of each; Nimblegen) was added to the 20 μl captured library (beads in H₂O), vortexed to mix and PCR amplified using the following PCR cycling conditions. Initial incubation at 98 °C for 45 s, 14 cycles of 98 °C for 15 s, 60 °C for 30 s and 72 °C for 30 s. Final incubation of 72 °C for 1 min and hold at 4 °C. Captured DNA was purified using 1.8:1 AMPure XP Beads: DNA ratio. Capture efficiency was determined to be between 120 and 240 fold using a Nimblegen Sequence Capture control locus qPCR assay. Neo-centromere captured DNA libraries were sized and quality controlled on a D1000 Tapestation tape (Agilent), and paired-end sequenced (50 bp) on an Illumina MiSeq (Edinburgh Genomics). Sequence read depth for HSA3 was 62 million mono nucleosome reads and 67 million di nucleosome reads. Sequence read depth for Neo3 was 26 million mono nucleosome reads and 31 million di nucleosome reads. These data are equivalent to mapping 42 000 nucleosomes per Kb in HSA3 and 17 000 nucleosomes per Kb in Neo3.

#### Nucleosome positioning analysis
Paired end sequencing reads were mapped to hg38 using Bowtie2 with high quality (mq > 20) and paired reads selected for further analysis. Start and end positions of reads were extracted from bamfiles using bedtools bamtobed function and analysed in R. The NucleR package was used to calculate nucleosomal dyad positions with 40 bp trimming and coverage, data was formatted in the ZOO package and plotted using the lattice package. The acf function in R was used to calculate nucleosome autocorrelation.

#### Copy number analysis
Genomic DNA libraries from the high passage (passage 100) Neo3 cells were generated using the NEBNext® Ultra™ II DNA Library Prep Kit for Illumina® (NEB #E7645L) following the manufacturer's instructions. Libraries were sized and quality controlled on a D1000 Tapestation tape (Agilent) and Illumina sequencing (paired-end DNA-seq of 50 bp read length) were performed on NovaSeq S1 (Edinburgh Genomics). FASTQ sequence files were obtained and the DNA-seq reads were aligned to a Human(hg38) /Hamster (GCA_003668045.1) hybrid reference genome using Bowtie2 and processed with Samtools v1.6[85], and the deepTools "bamCoverage" tool[86] with RPKM normalisation. Sequence read depth for Neo3 replicates was 166 million (9x coverage/base) and 118 million (5x coverage/base), with mapping efficiencies of 96 and 91%.

#### Reporting summary
Further information on research design is available in the Nature Research Reporting Summary linked to this article.

## Data availability
The data that support this study are available from the corresponding author upon reasonable request. The data reported in this paper are publicly available on GEO: SuperSeries GSE195886. This is composed of the following SubSeries: GSE195883 (Agilent 'open' chromatin-chip), GSE195884 (Agilent bTMP-chip), GSE195885 (Agilent ChIP-chip), GSE196155 (TTseq), and GSE196160 (RNAseq). Agilent arrays were designed using human reference genome (hg19) (GSE195883, GSE195884 and GSE195885). RNAseq data (GSE196160) is aligned to human reference genome (hg19). TTseq data (GSE196155) is aligned to Human(hg38) /Hamster (GCA_003668045.1) hybrid reference genome. Source data are provided with this paper.

## Code availability
Custom iVision (Version 4.5.6 r4) script used to calculate the distance between two fosmid probe signals is available upon request from the corresponding author.

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

## Acknowledgements

We would like to thank all members of our groups for useful discussions and to Alison Pidoux and Jim Allan for critical comments on the manuscript. This work was funded by the UK Medical Research Council (MR/J00913X/1; MC_UU_00007/13) (NG).

## Author contributions

C.N., C.H., C.R.C., A.B., R-S.N. and S.P. undertook experiments; C.N., G.R.G. and N.G. analysed data; M.R. and N.G. designed experiments; C.N. and N.G. wrote the manuscript with input from all authors.

## Competing interests

The authors declare no competing interests.
