## [Peer Review File · Nature Communications]

Editorial Note: This manuscript has been previously reviewed at another journal that is not operating a transparent peer review scheme. This document only contains reviewer comments and rebuttal letters for versions considered at Nature Communications

REVIEWERS' COMMENTS

Reviewer #1 (Remarks to the Author):

The authors have put a significant amount of effort addressing most of my comments and those of other reviewers- thank you.

While I am not fully persuaded the key findings of this ms are strictu sensu novel, the premise of this work is novel, as it compares the same DNA sequence in two epigenetically distinct contexts in the powerful human/hamster system. In this, they do a very good job extending key findings from native centromeres to a new system that can be used experimentally to dissect questions in a cleaner manner than what the rest of us have been attempting to do with native human centromeres which are repetitive, dynamically shifting, and hard to analyze despite Karen Miga & T2T's heroic tour-de-force filling of the genomic black hole.

More to the point, the work is convincing that the 3q neoCEN does adopt an open, RNAP2 transcriptionally permissible domain once CENP-A/C have assembled upon it. This is a key piece of evidence that will be critical for the study of quasi-neocentromere domains in the context of chromosome fragility arising at breakpoints in human cancer, and also for karyotype evolution, where centromeres shift over time.

In my view, these findings inform centromere biology and cancer biology fields, making it worthy of publication in Nat Comm.

Reviewer #2 (Remarks to the Author):

This is a revised manuscript from Naughton et al. addressing the chromatin structure induced by centromere formation. The manuscript takes a very powerful approach of deriving mouse:human hybrid cell lines containing the neocentromere chromosome to provide analysis of the neocentromere without complications of the presence of the homologous chromosome. The revised manuscript addresses most of the points raised in the initial review. However, one issue remains that I think is important for the impact of the manuscript. The revised manuscript provides further support that decompaction occurs at the neocentromere by including an analysis of Neo6 containing cell lines. This is an excellent inclusion, and I am not sure why these data are relegated to supplementary material. Analysis of the second Neocentromere 6 is critical to demonstrate these effects are due to the neocentromere, and not the process of creating the hybrid, which can lead to clonal selection (or idiosyncratic to this Neo3). This holds true for all the analysis. RNA pol II binding, histone PTMs, transcription and DNA supercoiling should all be analyzed on Neo6 (or independently derived clones of Neo3). Otherwise, it will be unclear if these are observations of this neocentromere derived mouse:human cell line, or general principles of neocentromere formation that will be of broad interest to the centromere community.

REVIEWERS' COMMENTS

Reviewer #1 (Remarks to the Author):

The authors have put a significant amount of effort addressing most of my comments and those of other reviewers- thank you.

While I am not fully persuaded the key findings of this ms are strictu sensu novel, the premise of this work is novel, as it compares the same DNA sequence in two epigenetically distinct contexts in the powerful human/hamster system. In this, they do a very good job extending key findings from native centromeres to a new system that can be used experimentally to dissect questions in a cleaner manner than what the rest of us have been attempting to do with native human centromeres which are repetitive, dynamically shifting, and hard to analyze despite Karen Miga & T2T's heroic tour-de-force filling of the genomic black hole.

More to the point, the work is convincing that the 3q neoCEN does adopt an open, RNAP2 transcriptionally permissible domain once CENP-A/C have assembled upon it. This is a key piece of evidence that will be critical for the study of quasi-neocentromere domains in the context of chromosome fragility arising at breakpoints in human cancer, and also for karyotype evolution, where centromeres shift over time.

In my view, these findings inform centromere biology and cancer biology fields, making it worthy of publication in Nat Comm.

We thank the reviewer for acknowledging the significant efforts we undertook in addressing their comments and those of the other reviewers. We are delighted that the reviewer considers our findings worthy of publication in Nat comm and thank them for their time and input.

Reviewer #2 (Remarks to the Author):

This is a revised manuscript from Naughton et al. addressing the chromatin structure induced by centromere formation. The manuscript takes a very powerful approach of deriving mouse:human hybrid cell lines containing the neocentromere chromosome to provide analysis of the neocentromere without complications of the presence of the homologous chromosome. The revised manuscript addresses most of the points raised in the initial review. However, one issue remains that I think is important for the impact of the manuscript. The revised manuscript provides further support that decompaction occurs at the neocentromere by including an analysis of Neo6 containing cell lines. This is an excellent inclusion, and I am not sure why these data are relegated to supplementary material. Analysis of the second Neocentromere 6 is critical to demonstrate these effects are due to the neocentromere, and not the process of creating the hybrid, which can lead to clonal selection (or idiosyncratic to this Neo3). This holds true for all the analysis. RNA pol II binding, histone PTMs, transcription and DNA supercoiling should all be analyzed on Neo6 (or independently derived clones of Neo3). Otherwise, it will be unclear if these are observations of this neocentromere derived mouse:human cell line, or general principles of neocentromere formation that will be of broad interest to the centromere community.

We agree with the reviewer that the data from the second neocentromere Neo6 is an excellent inclusion. We have added this data in revision and in supplementary as unfortunately there are no available human-hamster hybrid cell lines with this Neo6 centromere. This therefore prohibits further chromatin structural analysis of this neocentromere. We made several attempts to generate

these human-hamster hybrid cell lines but it seems the Neo6 neocentromere is unstable. All attempts to select this chromosome into a hamster hybrid resulted in either loss of the Neo6 chromosome or breakage of the chromosome at the neocentromere followed by fusion to a hamster chromosome. We have added the following sentence to the discussion, "Although chromatin fibre decompaction was also observed at a second neocentromere (Neo6) (Supplementary Fig. 6) lack of available human-hamster hybrid cell lines prohibited further analysis of this neocentromere." We thanks the reviewer for their time and helpful suggestions that have greatly improved our manuscript.